# Nanobiosensing Based on Electro-Optically Modulated Technology

**DOI:** 10.3390/nano13172400

**Published:** 2023-08-23

**Authors:** Shuang Li, Ziyue Qin, Jie Fu, Qiya Gao

**Affiliations:** Academy of Medical Engineering and Translational Medicine, Medical College, Tianjin University, Tianjin 300072, China; qinziyue@tju.edu.cn (Z.Q.); fujie314159@tju.edu.cn (J.F.); yehiagao_0326@tju.edu.cn (Q.G.)

**Keywords:** nanobiosensing, surface plasmon resonance, electrochemiluminescence, smartphone

## Abstract

At the nanoscale, metals exhibit special electrochemical and optical properties, which play an important role in nanobiosensing. In particular, surface plasmon resonance (SPR) based on precious metal nanoparticles, as a kind of tag-free biosensor technology, has brought high sensitivity, high reliability, and convenient operation to sensor detection. By applying an electrochemical excitation signal to the nanoplasma device, modulating its surface electron density, and realizing electrochemical coupling SPR, it can effectively complete the joint transmission of electrical and optical signals, increase the resonance shift of the spectrum, and further improve the sensitivity of the designed biosensor. In addition, smartphones are playing an increasingly important role in portable mobile sensor detection systems. These systems typically connect sensing devices to smartphones to perceive different types of information, from optical signals to electrochemical signals, providing ideas for the portability and low-cost design of these sensing systems. Among them, electrochemiluminescence (ECL), as a special electrochemically coupled optical technology, has good application prospects in mobile sensing detection due to its strong anti-interference ability, which is not affected by background light. In this review, the SPR is introduced using nanoparticles, and its response process is analyzed theoretically. Then, the mechanism and sensing application of electrochemistry coupled with SPR and ECL are emphatically introduced. Finally, it extends to the relevant research on electrochemically coupled optical sensing on mobile detection platforms.

## 1. Introduction

Metal nanoparticles were first applied to the Lycurgus cup in Rome in the 4th century AD. By embedding gold (Au) and silver (Ag) nanoparticles in glass, they can absorb and scatter specific regions of the visible light spectrum, presenting bright colors [1]. Due to the optical response of precious metal nanoparticles, a wine glass appears green when light shines from the outside and red when light shines from the inside. In this process, precious metal nanoparticles can convert the energy of incident photons into collective oscillations of surface electrons, resulting in wavelength selective absorption and high molar extinction coefficient light scattering. At the same time, the coherent oscillation of electrons on the surface of nanoparticles also generates significant electromagnetic field enhancement and radiation attenuation.

The interaction between light and noble metal nanoparticles generates a collective oscillation of conduction band electrons, which is called SPR. Only materials with a negative real dielectric constant and a positive virtual dielectric constant can better support surface plasmon polaritons, and the most commonly used precious metal nanoparticles are Au and Ag nanoparticles [2,3]. When the incident electromagnetic field matches the electromagnetic field of the oscillating electrons on the surface of the nanoparticles, resonance conditions can be met. This resonance oscillation causes a significant increase in the wavelength selectivity of the absorption, scattering, and electromagnetic fields on the surface of the nanoparticles [4,5]. In the process of SPR, the position of the formant of the absorption spectrum, scattering spectrum, and plasma extinction spectrum (absorption + scattering) of metal nanoparticles will show a corresponding red shift with the increase in the refractive index or the decrease in the distance between particles. Based on the above theory, if plasma nanoparticles are modified with a receptor (such as an antibody) that can specifically bind to the target analyte (such as an antigen), then the binding between the analyte and the receptor will lead to an increase in the refractive index around the nanoparticles or a decrease in particle spacing, which will lead to a red shift in the peak position of the plasma extinction spectrum. The concentration of the analyte or the degree of binding reaction can be determined based on the shift of the spectral peak position. Therefore, SPR sensors are widely used in chemical and biological sensing [6,7,8].

In order to understand the factors affecting the increase in absorption and scattering during plasma resonance, the Mie theory was proposed. Mie theory is the analytical expression of Maxwell equations with spherical boundary conditions that is used to describe the extinction spectrum of a given nanoparticle. In order to more accurately calculate the dielectric constant at different wavelength values and extend this theory to more complex nanoparticle shapes, a modified wavelength form of Mie theory was proposed [9,10,11]. As shown in Formula (1), *R* represents the radius of the particle, *ε_m_* is the dielectric constant of the surrounding medium, and *N* represents the electron density, *λ* represents the wavelength of incident light, *ε = ε_r_ + iε_i_* refers to the complex dielectric constant of a metal, *χ* determined by particle shape. Therefore, factors such as the shape of nanoparticles, the wavelength of incident light, the type of material, and the surrounding medium can all affect the absorption and scattering processes.
(1)$$C_{ext}=\frac{24\pi^{2}R^{3}\varepsilon_{m}^{\frac{3}{2}}N}{\lambda \ln\left(10\right)}\frac{\varepsilon_{i}}{{\left(\varepsilon_{r}+\chi \varepsilon_{m}\right)}^{2}+\varepsilon_{i}^{2}}$$

In the field of biosensors, when biological analytes are combined with the surface of nanoparticles, the refractive index of the surface of the nanoparticles will change, thus changing the movement of the SPR peak position. Among them, Ag nanoparticles have the highest negative dielectric constant among all plasma materials and are the most sensitive to changes in local refractive index. In addition, asymmetrically shaped nanoparticles are more sensitive to changes in the surface binding of biomolecules than spherical plasmas. The relationship between the position of the SPR peak and the binding of the adsorbent is described in Formula (2), where *m* is the refractive index sensitivity, Δ*n* is the refractive index change caused by the adsorbent. *d* refers to the thickness of the effective adsorption layer, and *l_d_* refers to the attenuation length of the electromagnetic field.
(2)$$\Delta \lambda =m(\Delta n)[1-\exp(\frac{-2d}{l_{d}})]$$

The refractive index sensitivity is usually obtained by calculating the slope of the SPR frequency and refractive index curve under a small range of refractive index using the Drude model of electronic structure [12,13]. Research shows that the influence of the electromagnetic field attenuation length on the SPR peak shift is mainly due to sensitivity to the shape of nanoparticles. The electromagnetic field attenuation length of many nanoparticles is similar to that of proteins in size (5~10 nm), which enables nanoparticles to sensitively sense the binding of biomolecules to their surfaces [14,15,16,17]. Therefore, the two key variables that determine the shift of the plasma formant are the refractive index difference of the absorbance relative to the solution (Δ*n*) and the degree of surface binding between the analytes and the nanoparticles (*d*). Due to the sensitivity of the refractive index being the key to detecting target biomolecules, many studies in recent years have attempted to make the nanoparticle matrix highly sensitive to changes in refractive index. The relevant simulation results show that when the SPR peak of the material moves from the blue wavelength to the red wavelength, the sensitivity of the refractive index increases in a linear way [18,19,20]. By creating larger and/or asymmetric nanoparticles, SPR peaks can be raised to longer wavelengths. In addition, when two plasmas approach each other, the electromagnetic field between them will generate a certain degree of mutual coupling, which can also cause a significant wavelength shift [21,22]. Because the presence of nearby plasma-active substances often has a greater impact on SPR, many sensing studies use plasma coupling as a technical means to improve the response of detection signals.

## 2. Electrochemical Coupled SPR

As people pay more and more attention to the application of nanostructures in charge transfer and storage, it is crucial to control the charge transfer at the nanoscale through the physical and chemical processes in the reaction [23,24,25]. Drude describes metals as plasmas of free electrons, which form the physical basis for their optical and electrical properties. For some metals, quantum effects are ignored when the size exceeds 10 nm, and the classic Drude model can provide quantitative detection results in optical and electrical systems. However, in nanometers, structurally, the stimulated luminescence of plasma and the influence of the environment have a certain impact on conductivity, which has a high surface sensitivity; that is, they have a strong response to environmental changes that occur on the surface, leading to the development of new unlabeled biosensors. Typically, these sensors do not require sensitive labels, such as fluorescent groups, radioactive labels, or quantum dots, and are mainly based on electrical or optical signal transduction [26,27]. On this basis, electrochemically coupled optical technology has been developed, and with the development of nanomaterials, electrochemically coupled optical detection has become more inclined towards simplification of operation, stable curing of testing, and miniaturization of systems.

### 2.1. Coupling Mechanism

SPR is an optical phenomenon generated by the collective movement of electrons on the surface of metal nanoparticles, and the existence of an electric field will cause the deposition of electroactive molecules on the surface of metal nanoparticles, which will lead to the movement of the plasma formant [28,29,30,31]. Therefore, when the electrochemical signal acts on the plasma nanoparticles, the positive displacement of the potential reduces the density of free electrons in the nanoparticles, leading to a red shift in the peak position of the extinction spectrum, resulting in the electrochemical coupling of the SPR effect. Figure 1a shows the relationship between the wavelength, potential, and extinction spectra of plasma nanoparticles. The two curves in Figure 1b (wavelength extinction spectrum) are the cross-sectional views of Figure 1a at potentials *E*_1_ and *E*_2_. It can be clearly seen that when the potential increases between *E*_1_ and *E*_2_, the peak position changes from *λ*_1_*** move to *λ*_2_***. On the other hand, Figure 1c (potential extinction spectrum) shows that at a wavelength of *λ*_3_, a new formant can be obtained in the extinction spectrum through potential scanning, and the peak position voltage is *E*_3_***. The optical properties of plasma nanoparticles also depend on the refractive index changes of the surrounding medium, resulting in a red shift at the peak position in the wavelength extinction spectrum as the refractive index increases (Figure 1d). Figure 1e shows a cross-sectional view at a potential equal to *E_c_*, showing the peak position from *n_A_* to *n_B_* as the refractive index increases *λ_A_** move to *λ_B_**. Meanwhile, Figure 1f shows that at wavelengths of *λ_D_*, the cross-sectional view at D shows that as the refractive index increases from *n_A_* to *n_B_*, the peak potential shifts from *E_A_** negative to *E_B_**. Overall, the increase in refractive index change *n_B_ − n_A_* will lead to an increase in negative potential shift −(*E_B_** − *E_A_**) and a positive wavelength shift of *λ_B_* − λ_A_*.*

The relationship between applied potential *E = E*_0_
*+* Δ*E* and wavelength is given by Formula (3), where *C*, *V*, *F*, and *d* are the capacitance, molar volume, faraday constant, and diameter of metal nanoparticles, respectively. It is reported that the *C* value of the citrate-coated Au nanoparticles with *d* = 3.7~40.8 nm is about 70 μF/cm^2^ when compared with the Ag/Ag chloride/saturated potassium chloride electrode (at the potential from −0.4 V to +0.6 V) [32]. The *V* value of Au nanoparticles is 10.2 cm^3^/mol. Within the range of the actual refractive index, the peak potential *E_λ_** is almost linear and depends on the change in refractive index. It can be seen from the slope of the approximate line that the theoretical refractive index sensitivity *S_E-n_* of the electrochemical coupling SPR sensor for Au nanoparticles with a diameter of 13 nm is about 10.3 V/RIU. Both electrochemistry and SPR realize detection by sensing the relative change of the reaction interface. Due to the potential sensitivity of noble metal plasma surfaces, electrochemical signals can be introduced into SPR to realize electro-optic coupling applications.
(3)$$\frac{\lambda_{E}^{*2}}{\lambda_{E0}^{*2}}-1=\frac{6CV\Delta E}{Fd}$$

### 2.2. Sensing Application

Continuous, sensitive, and real-time analysis of complex, unknown samples is a fundamental requirement widely used in biosensing detection. Although many existing biosensing technologies have shown extremely high sensitivity in laboratory environments, they are still subject to interference from high-strength non-specific reactions in complex samples in practical applications due to the hybrid effect caused by non-specific adsorption on the sensor surface [33]. A potential solution to address this challenge is to integrate compatible biosensing methods into an analytical tool that can measure and analyze more supplementary parameters. The integration of electrochemistry and SPR sensing technology provides a very attractive method for converting biochemical response results into a variety of measurable analytical signals.

Research shows that the redox current in the electrochemical reaction can cause a shift in the resonance wavelength in the spectrum to increase the SPR detection signal. Basing on cyclic voltammetry coupled with SPR technology, Zhang et al. realized the highly sensitive detection of bovine serum albumin (BSA), which used a mixture of nanocups and nanoparticles to produce a hybrid structure called nanoscale cup arrays (nanoCA) for electrochemical enhancement of LSPR [34]. As shown in Figure 2a, the oxidation reduction potential of BSA appears to be 0.6 V. With the increase in BSA concentration, the reduction current increases, which further causes the movement of the SPR wave valley and enhances the signal of BSA detection. Figure 2b shows the transmission spectrum of nanoCA with CV scanning. Compared with PBS, without CV scanning, the LSPR shift signals of BSA decreased by only 6 nm, while with CV scanning, the LSPR shift signals increased to 13 nm due to the combined effect of electrochemical current from the redox peak. Li et al. activated dopamine itself to generate redox current by cyclic voltammetry and simultaneously recorded the change of electrode surface refractive index using a spectrum, which can achieve highly sensitive and specific dopamine detection [35]. As shown in Figure 2b, the arrow indicates the oxidation current position of dopamine, with the first oxidation peak occurring at 55 s and an interval of 280 s from the next oxidation peak. In order to better observe the changes in the intensity and peak position of the transmission spectrum during cyclic voltammetry scanning, the color contour line map calibrates and presents each transmission spectrum as a time function. The intensity of the transmission spectrum fluctuates periodically with the appearance of the dopamine oxidation current peak and reaches the maximum of the spectral intensity at the current peak. The peak position shows an obvious periodic red shift.

In addition, anodic stripping voltammetry can determine multiple metal ions at once by combining voltammetry technology with constant potential electrolytic enrichment. However, due to the interference of background current, the oxidation and reduction of oxygen and the precipitation of hydrogen generate additional current, which limits the detection of trace metal ions. The interference of background current can be effectively removed through the coupling of anodic stripping voltammetry and SPR, which greatly improves the sensitivity and signal-to-noise ratio of metal ion detection [36]. As shown in the mixed detection of zinc, lead, and copper in Figure 3a, there is not much change in the spectral transmittance during the dissolution stage. Since the difference in transmissivity reflects the change rate of the number of electroplating metal nanoparticles remaining on the device surface, which is related to the dissolution current in anodic stripping voltammetry detection, the first-order difference between transmissivity and time can be made through the difference between every two sampling data points in the transmissivity time curve, which can better realize the distinction of metal ions.

In our previous work, we constructed an electrophoretic-enhanced SPR sensing system. The Au nanocup array is used as an electro-optic sensor, and the transparent indium tin oxide electrode is used as a positive electrode [37]. Direct current is applied between the Au nanocup array sensor and the indium tin oxide electrode. As shown in Figure 3b, using the “sandwich sandwich” strategy, polyethylene glycol with thiol groups is fixed on the surface of the Au nanocup array device through Au sulfur bonds, forming a good biosensitive film. Then, the thrombin specific shear peptide is fixed on the polyethylene glycol biosensitive membrane through amide action. Finally, the end of the peptide chain was modified with negatively charged BSA. When thrombin is introduced onto the surface of the Au nanocup array sensor, the peptide cleavage reaction of thrombin is carried out at a specific position of the polypeptide. With the help of the electric field between the indium tin oxide electrode and the Au nanocup array sensor, the shear peptide and electronegativity of BSA can be separated. Due to the change in the refractive index of the sensor surface, the formant in the transmission spectrum shifted, and the construction of the sensor for determining the catalytic activity of thrombin was completed. In addition, we also modified Au nanoparticles and Ag nanoparticles on the nanocone array by electron beam evaporation and electrochemical reduction deposition, and then carried out linear scanning voltammetry and SPR sensing on the Au/Ag nanocone array sensor to record dynamic electro-optic spectrum signals for sialic acid detection (Figure 3c) [38]. The charge distribution induced by plasma is concentrated on the surface of metal nanoparticles, leading to the excitation of hot electrons and hot holes. If the electron energy exceeds the Fermi energy level of the material, the photoexcited electrons will generate light emission through electron-electron or electron-phonon collisions. Then light absorption and SPR will be generated on the nanosensor, accompanied by electromagnetic attenuation on the femtosecond scale, and energy will be transferred to hot electrons in a non-radiative manner.

Due to the high requirement for relative balance between electrodes and electrochemical measurements, it is often necessary to keep the analyte stationary as much as possible during detection. Any relative movement between the analyte solution and the surface of the sensing electrode will cause severe fluctuations in the voltammetry curve. Baseline drift during the detection process may cause vertical fluctuations in the potential, while impurities in the analyte (such as buffer solutions and dissolved gases) may also cause horizontal fluctuations in the potential. However, the use of SPR technology can achieve real-time analysis of the target under dynamic conditions, which greatly improves the robustness of the detection. Therefore, electrochemically coupled SPR detection has a strong anti-interference ability against the impact of temperature, buffer solution, and analogues and has a good application prospect in dynamic detection conditions with high integration, such as the miniaturized mobile sensing platform.

## 3. ECL

ECL is a reaction on the electrode surface under the condition of an electrolytic cell that is mainly controlled by the change of electrode potential and emits light signals of a specific wavelength with the process of excited state electron transfer [37,38]. In a sense, it is an ideal combination of electrochemistry and optics because it not only maintains the high sensitivity and wide dynamic range of traditional chemiluminescence but also demonstrates the simplicity, stability, and convenience of electrochemical methods. As a light emission technology, ECL has unique advantages compared to other luminescence methods such as photoluminescence and chemiluminescence. Especially compared to chemiluminescence, ECL has superior temporal and spatially precise control characteristics for light emission, which can significantly improve the signal-to-noise ratio in the detection process. In addition, compared to photoluminescence, which needs to withstand the influence of non-selective photoexcitation induced background, ECL does not require the use of external light sources and has zero background interference characteristics, avoiding disturbance caused by light scattering. Therefore, ECL has become a powerful analytical technique, often used for sensing the detection of trace target analytes, and plays an important role in fields such as medical diagnosis, food safety, and biochemical warfare agent detection [39,40,41].

### 3.1. Luminescence Mechanism

ECL is generated by the recombination of electrogenerated free radicals, and its mechanism can be divided into two categories based on the source of free radicals: the annihilation mechanism and the co-reactant mechanism. For the former, free radicals are generated by a single emitter, while the latter involves a set of bimolecular electrochemical reactions between the luminescent body and the co-reactant.

(1)Annihilation mechanism

The annihilation reaction involves the formation of electrochemical intermediates on the electrode surface, and then the intermediates interact with each other and undergo the formation of a ground state and an excited state. Finally, the excited state emits light due to relaxation. A typical example is 9,10-biphenylanthracene [42,43]. Diphenylanthracene (DPA) undergoes oxidation (Formula (4)) and reduction (Formula (5)) reactions on the electrode surface, respectively. Then, the oxidation intermediate (DPA^+^) and the reduction intermediate (DPA^−^) undergo an annihilation reaction and generate an excited state (DPA*) (Formula (6)). The last unstable excited state (DPA*) generates a luminous signal (Formula (7)) in the process of returning to the ground state (DPA).
(4)$$\mathrm{DPA}-e^{-}\to {\mathrm{DPA}}^{+\cdot }$$
(5)$$\mathrm{DPA}+e^{+}\to {\mathrm{DPA}}^{-\cdot }$$
(6)$${\mathrm{DPA}}^{-\cdot }+{\mathrm{DPA}}^{+\cdot }\to \mathrm{DPA}+{\mathrm{DPA}}^{*}$$
(7)$${\mathrm{DPA}}^{*}\to \mathrm{DPA}+hv$$

Gibbs free energy related to the annihilation process is obtained by calculating the oxidation and reduction potentials of Formulas (4) and (5), as shown in Formula (8), where Δ*G* represents Gibbs free energy, *F* is the Faraday constant, and $E_{reduction}^{0}$ and $E_{oxidation}^{0}$ are reduction potential and oxidation potential, respectively. Formula (9) provides the calculation formula for Gibbs’ free energy-related enthalpy.
(8)$$\Delta G=-nF(E_{reduction}^{0}-E_{oxidation}^{0})$$
(9)ΔG=ΔH−TΔS

If the enthalpy exceeds the minimum value of energy required from the ground state to the excited state, the reaction is defined as “energy sufficient” or follows the “singlet state,” and biphenylanthracene is a reaction conforming to the “singlet state”. On the contrary, if the enthalpy is lower than the minimum value of the energy required to move from the ground state to the excited state but still exceeds the triplet state energy, triplet state-triplet state annihilation will occur. A typical example of triplet state annihilation in ECL is ruthenium bipyridine derivatives. In addition, the annihilation reaction can also form excited dimers and excited complexes. In this case, the main advantage of the annihilation process is that it only requires luminescent substances, solvents, and supporting electrolytes to produce luminescence.

(2)Co-reactant mechanism

Co-reactant ECL is usually generated by applying an electric potential to the electrode, and the luminescent material is either deposited on the electrode surface or in a solution system, accompanied by the presence of a co-reactant, resulting in the luminescence phenomenon. When a potential (positive or negative) is applied to the electrode, the luminescent material and the co-reactant undergo simultaneous oxidation or reduction reactions, forming free radicals and intermediates. Then, the intermediate state will interact with the oxidized or reduced emitters to generate the excited state with the formation of highly active oxidized or reduced substances, and then the excited state will emit light signals when returning to the steady state. In typical ECL reactions based on ruthenium bipyridyl emitters, as shown in Formulas (10)–(14), Tripropylamine (TPA) is the main co-reagent in the process of co-reactant luminescence. Through electron transfer or chemical reaction, tripropylamine deprotonation product (TPA^•^) reacts with the emitter to generate an excited state (*R**), and the unstable excited state (*R**) is generated along with the luminescence signal in the process of returning to the ground state (*R*).
(10)$$R-e^{-}\to R^{+\cdot }$$
(11)$$\mathrm{TPA}-e^{-}\to {\mathrm{TPA}}^{+\cdot }$$
(12)$${\mathrm{TPA}}^{+\cdot }\to {\mathrm{TPA}}^{\cdot }+H^{+}$$
(13)$$R^{+\cdot }+{\mathrm{TPA}}^{\cdot }\to R^{*}$$
(14)$$R^{*}\to R+hv$$

Other co-reactants include oxalate and pyruvic acid salt ions operating in the redox mode, and persulfate, hydrazine, and hydrogen peroxide operating in the redox mode. Oxidation-reduction type ECL usually occurs under positive potential conditions, and the release reaction of oxygen in aqueous solution is slow, with little impact on ECL [44]. On the contrary, many reduction oxidation types of ECL require very negative potential conditions, which generate a large amount of hydrogen gas and cause immediate decomposition of electrogenerated intermediates, making it difficult to achieve stable reduction oxidation types of ECL in a liquid-phase environment [45]. Therefore, ECL based on co-reactants is particularly useful when it is necessary to avoid oxygen dissolution, which is of great significance for the detection of analytical samples without deoxygenation. It should be noted that as long as the electrolytic cell keeps working, the light-emitting body can be recycled near the electrode, and the co-reagent is constantly consumed in the whole reaction process, so it is only necessary to ensure that there is sufficient co-reagent in the reaction system during the reaction process to ensure continuous and stable luminescence. Appropriate co-reactants can easily oxidize or reduce and then undergo a rapid chemical reaction to form an intermediate with sufficient oxidation or reduction ability to stimulate the formation of the excited state of the emitter, thus promoting the generation of luminescence.

### 3.2. Luminescent Materials and Applications

Electrochemical luminescent materials can generally be divided into three categories: inorganic, organic, and nanomaterials. The abundant luminescent materials play a very important role in the development of ECL [46,47]. Specifically, ruthenium tripyridine (Ru(bpy)_3_^2+^), as an inorganic emitter and the most successful emitter, has a very wide range of applications, mainly due to its strong luminescence and solubility in aqueous or non-aqueous solvents and its reversible electron transfer ability at easily available potential [48]. Considering the electrochemical and spectroscopic properties required for ECL exhibited by many metal complexes, such as iridium (Ir), Au, Ag, copper (Cu), platinum (Pt), aluminum (Al), cadmium (Cd), chromium (Cr), molybdenum (Mo), tungsten (W), europium (Eu), osmium (Os), palladium (Pd), thallium (Tl), rhenium (Re), terbium (Tb), silicon (Si), and mercury (Hg) have also been reported for ECL detection [49].

In organic systems, diphenylanthracene and its derivatives have attracted widespread attention in the field of ECL due to their significant luminescent properties. In addition, studies have found that fluorene, thiophene triazole, and their derivatives can also generate ECL reactions [50,51]. The optical, electrical, electrochemical, and luminescent properties of nanoparticles (including polymer and metal nanoparticles) make them an attractive material. Among them, silica nanoparticles have proven to be good electrochemical luminescent materials due to their easy surface chemical properties for modification and functionalization [52].

Subsequently, semiconductor nanocrystals, or quantum dots, received widespread attention due to their excellent luminescent properties and applications in many important fields [53,54,55]. Quantum dots have a high fluorescence quantum yield, stability against photobleaching, and size-controlled luminescence characteristics, which make them a new nanomaterial used for sensing analysis [56,57,58]. Among them, ECL research first reported that silicon nanocrystalline quantum dots were used [59], and then various quantum dots, including cadmium sulfide (CdS), cadmium selenide (CdSe), cadmium telluride (CdTe), zinc sulfide (ZnS), and silver selenide (Ag_2_Se), were also used for ECL detection [60]. In addition, other hybrid nanomaterials with various compositions, sizes, and shapes, such as metal nanoclusters [61], carbon nanodots [62], carbon nitride [63], and graphene and its composites [64], are also used as ECL materials.

Due to its inherent sensitivity, negligible background, and simple controllability, ECL has been recognized as a powerful analytical technique. In the past few decades, various bioassay methods have been used for ECL detection of target analytes. For the construction of sensitive ECL sensing detection methods, they can generally be divided into five categories [65]. Firstly, the analyte can inhibit or enhance the ECL reaction by means of energy transfer or electron transfer. Secondly, through redox or surface bonding/detachment, the enhancement or decomposition of the ECL is realized. Thirdly, by producing or consuming co-reactant through an enzyme-linked immunosorbent reaction, changes in ECL signals can be achieved. Fourthly, an ECL sensing strategy for signal cutoff is achieved through the spatial hindrance caused by biological cognitive reactions or target-induced deposition. Fifthly, an efficient ECL resonance energy transfer sensing strategy is achieved by overlapping the spectra of the receptor and donor. Based on these methods, ECL technology is widely used in metal ion detection [66,67], immunoassay [68], gene sensing [69,70,71], early cancer diagnosis [72], and other fields.

### 3.3. ECL for POCT

Due to the urgent need for rapid diagnosis, point-of-care testing (POCT) has been proposed and has attracted widespread attention, mainly using portable, low-cost, and user-friendly devices to provide rapid analysis and real-time monitoring of health status [73,74,75]. Research has shown that the design of real-time detection systems based on ECL technology mainly includes two key components: ECL excitation and luminescent detectors.

(1)ECL excitation

In traditional ECL analysis, the luminescence signal is excited and controlled by an electrochemical workstation. Due to the high cost of electrochemical workstations, developing alternative ECL excitations for real-time detection is a top priority. Among them, portable and low-cost rechargeable batteries, due to their advantages of high efficiency, stable voltage output, large capacity, and low self-discharge rate, have become a good choice for electrical excitation in the manufacturing of portable electrochemical luminescent devices for real-time detection (Figure 4a,b) [76,77]. In addition, as shown in Figure 4c, rechargeable supercapacitors have also been reported by Kadimisetty et al. to provide voltage excitation for ECL [78]. The supercapacitors use solar energy for rapid power supply, and the developed ECL protein immunoarray can detect three cancer biomarkers in serum within 35 min, which is conducive to on-site diagnosis in resource-limited environments. In addition to charging batteries, Delaney et al. proposed an ECL excitation method that does not require the use of a potentiostat [79]. As shown in Figure 4d,e, voltage excitation comes from the audio port of the smartphone and is modulated through the audio output to obtain a square wave signal with a positive interval of 0.1 s and a negative interval of 0.04 s, which is used to stimulate the generation of ECL signals. Afterwards, they further designed a novel and universal ECL excitation based on the standard USB On-The-Go (USB-OTG) specification [80]. As shown in Figure 4f, the luminescence excitation voltage can be driven from any USB-OTG certified smartphone USB port and modulated through an audio interface to be output to the ECL reaction cell.

In our previous work, we used graphene quantum dots/Ag nanoparticles to amplify and enhance the ECL signal, combining it with the ECL system based on smartphones to detect Escherichia coli (Figure 5a). In the ECL system, tripyridine ruthenium is used as the luminescent body, tripropylamine as the coreactant, and graphene quantum dots/Ag nanoparticles as the enhancer to stabilize and enhance the luminescence response. Using the universal serial bus USB-OTG on the smartphone to provide voltage excitation, the camera captures light and achieves concentration detection of Escherichia coli from 10 cfu/mL to 10^7^ cfu/mL [81]. In addition, we developed a silica nanopore enhanced ECL system on a smartphone, using the universal serial bus USB-OTG as the electrical stimulus and the camera to capture the luminescence (Figure 5b). Using positively charged tripyridine ruthenium as the luminescent material in the ECL reaction, the signal amplification in the ECL detection was achieved by enhancing the luminescence signal response of the nanopores with negative charge characteristics on the inner wall of the pores [82]. The smartphone detection system also uses inexpensive, flexible, and disposable screen-printed electrodes, which is convenient for on-site biochemical detection applications.

To further simplify the design of ECL excitation, a portable, thermally powered ECL visualization sensor was proposed by Hao et al. [83]. The sensor consists of a micropower supply and an easily prepared transistor array (Figure 5c). The unique power supply structure provides a valuable reference for the miniaturization of electrochemical light-emitting devices and facilitates subsequent on-site operation and real-time detection. On this basis, Zhang et al. developed a stable, environmentally friendly, metal-free, self-powered three-dimensional microfluidics ECL biosensor platform based on the origami principle [84]. As shown in Figure 5d, the platform assembles the energy part and the sensing part on a three-dimensional paper chip. The microfluidic origami ECL device has broad application prospects in portable, green, low-cost, and disposable detection equipment.

(2)Luminous detector

The collection of luminescent signals is another important component of electrochemical luminescent sensing systems. The traditional luminous signal acquisition is completed by the Photomultiplier tube, mainly because of its high sensitivity. With the increasing demand for portable and user-friendly ECL detectors in real-time detection, a number of alternative optical sensors have emerged, such as charge-coupled devices (CCD), complementary metal oxide semiconductor devices (CMOS), and photodiodes [85]. In order to reduce testing costs, numerous studies have reported the use of digital cameras as electrochemical luminescent detectors. As shown in Figure 6a, Doeven et al. utilized the excitation and emission characteristics of ECL materials, combined with the inherent color selectivity of traditional digital cameras, to effectively distinguish red, green, and blue emitters in the three-dimensional space of ECL intensity, applied potential, and emission wavelength, creating a new multi-channel ECL detection strategy suitable for developing low-cost and portable clinical diagnostic equipment [86]. In addition, based on a luminol/hydrogen peroxide system on a bipolar electrode array platform and using digital cameras to visualize ECL signals, Khoshfetrat et al. developed a radio luminescence DNA array for the visual genotyping of different single-nucleotide polymorphisms (SNPs) (Figure 6b) [87]. In detail, DNA probes are used to modify the array’s anodic poles and identify targets, and then the array is exposed to different single-base modified luminol-platinum nanoparticles for SNP genotype. This biosensor can detect thermodynamically stable SNPs in the range of 2–600 pM. Combining the advantages of a bipolar electrode with the high visual sensitivity of ECL, it has great potential for achieving sensitive screening of different SNPs.

Smartphones, due to their high imaging and computing power as well as open-source operating systems, are playing an increasingly crucial role in sensor analysis and medical monitoring [90]. The huge user base of smartphones has further promoted the rapid development of embedded hardware and software, as well as high-end imaging and sensing technologies in mobile phones. This has gradually made smartphones a promising intelligent platform for developing various biological sensor analysis devices that can be used for fast, real-time, and on-site detection, greatly simplifying design and reducing the cost of detection systems [91,92]. The portability and ubiquitous availability of smartphones provide great convenience for the development of portable electrochemical luminescent devices. As shown in Figure 6c, by combining inkjet-printed paper-based microfluidic electrodes with a smartphone-based ECL detection system, a simple and inexpensive sensing analysis platform can be constructed without traditional photodetectors [88]. Based on the mechanism of orange luminescence caused by the ECL reaction of tris(2,2′-bipyridyl)ruthenium(II) with certain analytes, Delaney et al. analyzed the intensity of red pixels in digital images emitted by ECL. The constructed calibration curve indicates that smartphones can detect 250 μM DBAE.

In addition, Chen et al. proposed a handheld bipolar ECL system that uses rechargeable batteries and smartphones to read ECL signals (Figure 6d) [89]. The system is based on the ECL reaction of luminol/H_2_O_2_ for quantitative detection and is suitable for glucose detection in PBS and artificial urine. It has advantages such as high sensitivity and stability and has good development prospects in fields such as POCT, health monitoring, and environmental monitoring. Additionally, the development of bipolar electrochemistry, microfluidics chips, wireless power supply and data transmission, as well as optical detection technology, has made the performance of ECL sensors unprecedentedly improved for real-time detection applications. Undoubtedly, the significant progress and breakthroughs in real-time detection in the coming years will also greatly promote the development of these emerging technologies. Based on the above progress, it can be found that integrating various processes of ECL (including sample collection, separation, and detection) into an “all-in-one machine” while cleverly combining different technologies and elements to manufacture integrated ECL sensors and ensuring the high specificity, sensitivity, and repeatability of sensors in real-time detection is still a direction that needs continuous efforts and exploration.

## 4. Prospect

Although this article has conducted relevant research on nanobiosensing motion detection based on electrochemically coupled optical technology, there are still some problems that need to be solved to further promote the application of electrochemically coupled optical technology in biosensing motion detection. First, the strictness of anti-interference requirements limited to optical sensing has greatly increased the complexity of optical path construction, limiting the realization of electrochemical coupling SPR resonance on smartphones. In addition, currently, nanoarray devices only achieve single biological modifications, only utilizing the improvement in sensing sensitivity brought by their nanostructures, and do not fully utilize the advantages of their array structures. Further exploration is needed to achieve multi-channel detection through the modification of different sensitive layers, enriching the output signals of nanoarray sensors. Secondly, this review has demonstrated the excellent electrochemical/optical sensing properties of nanomaterials. Then the biosensor characteristics of metal carbide nanomaterials with a two-dimensional layered structure, such as graphite oxide derivatives and MXene, need to be further studied to broaden the types of detection probes and promote the development of biosensors. However, the limited sensing characteristics brought by nanomaterials cannot fully meet the demand for biosensing specific detection. Therefore, further research is needed to improve the sensing selectivity of nanomaterials by coupling them with biologically sensitive layers such as proteins, peptides, or DNA. Finally, this study conducted a series of detailed studies on ECL detection based on smartphones. The sensing system is sufficiently complete to meet the relevant biological sensing and detection needs; however, there are still many aspects that can be improved with the development of ECL technology. Although the luminescent material is not consumed in the reaction, it is difficult to recycle and reuse. Solid state luminescence not only facilitates detection operations but also reduces testing costs by fixing and modifying the luminescent material on the electrode surface, which is conducive to mobile detection applications. Although current research utilizes mobile phone cameras to dynamically capture luminous videos, the luminous processing is still based on luminous imaging and does not directly analyze the optical signal. Therefore, real-time processing of luminous signals using smartphone light sensors and other components is another area that needs improvement. In the future, there is still a long way to go for related in-depth research, such as integrating more precise sensors on smartphones or developing more precise detection technologies to achieve quantitative analysis of fingerprint substances and promote mobile biosensing detection during the unlocking process of mobile phones. Given the excellent performance of ECL in imaging analysis, further exploration can be made in the spatial distribution of luminescence, such as multicolor luminescence. At the same time, the convenience of image processing using a smartphone can enable multi-channel biosensing detection applications on multicolor luminescent materials.

The ideas for future work are:(1)Research on multi-channel biosensing detection based on nanoarray sensors. Precious metal nanoarray sensors were prepared to realize electrochemical coupling SPR detection and modify multiple biosensitive layers on them. Build a multi-channel electrochemically coupled optical detection platform based on nanoarray sensors. Realize simultaneous detection of multiple target analytes.(2)Research on sensing detection based on a nanomaterial composite biosensitive layer studies the electrochemical/optical sensing characteristics of metal carbide materials with a two-dimensional layered structure. Preparation of composite probes for nanomaterials and biosensitive layers to achieve specific biosensing detection of target analytes.(3)Research on biosensing detection based on quantum dots in solid state ECL built a dynamic real-time ECL analysis system based on smartphone light sensors. And on this system, study the ECL-related characteristics of new quantum dot nanomaterials and complete the construction of solid-state ECL sensing components. Implement relevant biosensing detection applications.(4)Research on multicolor luminescent biosensing detection based on smartphone preparation of potential resolved multicolor ECL-sensitive probes and construction of multi-channel biosensing components. Build an ECL platform based on smartphones and conduct multimodal analysis of luminescence signals. Furthermore, mobile sensing for the detection of complex biological samples can be achieved.

## 5. Conclusions

This paper mainly introduces the basic concepts of local SPR, electrochemically coupled local SPR resonance, ECL, and related biosensor detection applications. Firstly, the local SPR is induced by nanoparticles, and its sensing mechanism is briefly introduced. Then the role of electrochemistry in improving local SPR signals is introduced, and the mechanism of electrochemical coupling of local SPR and its applications in biosensors are emphatically introduced. In order to meet the growing demand for mobile detection, ECL technology is introduced, and its luminescence mechanism, luminescent materials, and applications are described in detail. Finally, from the perspective of real-time detection applications, a detailed analysis was conducted on the miniaturization and intelligent development of ECL excitation and luminescent detectors.

## Figures and Tables

**Figure 1 nanomaterials-13-02400-f001:**
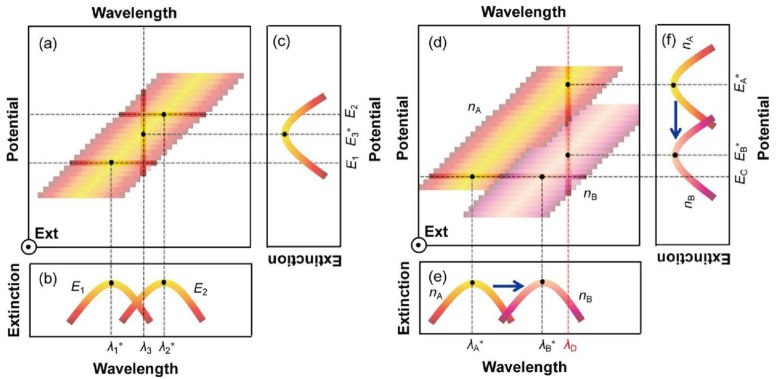
Wavelength Potential Extinction Spectra: (**a**) The relationship between the wavelength, potential, and extinction spectra of plasma nanoparticles. (**b**) The cross-section at *E*_1_ and *E*_2_. (**c**) The cross-section at *λ*_3_. (**d**) The relationship between the corresponding wavelength, potential, and extinction spectrum when the refractive index is *n_A_* and *n_B_*. (**e**) The cross-section at *E_c_*. (**f**) The cross-section at *λ_D_*. [31].

**Figure 2 nanomaterials-13-02400-f002:**
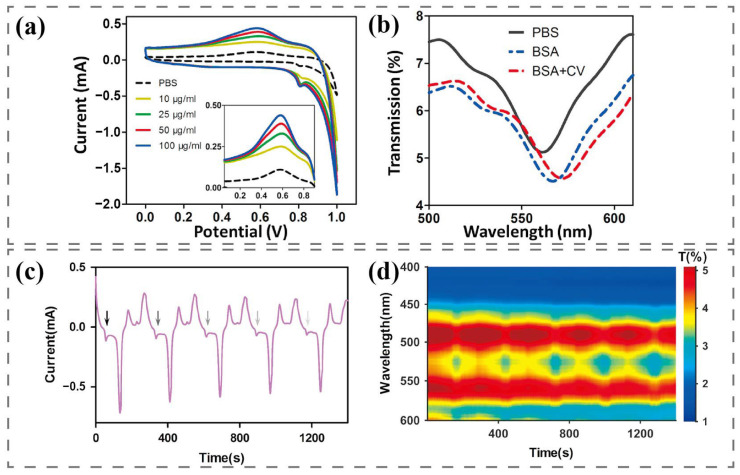
Electrochemically coupled SPR sensing applications: (**a**) Electrochemical CV scanning for BSA detection. (**b**) Transmission spectra of nanoCA for BSA detection [34]. (**c**) Dopamine detection: CV current unraveled in the time domain. (**d**) Dopamine detection: spectral responses of the nanosensor as a function of scanning time [35].

**Figure 3 nanomaterials-13-02400-f003:**
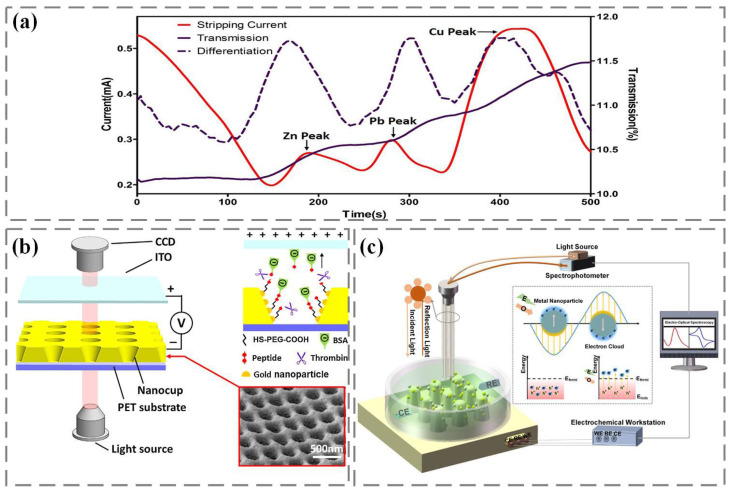
Electrochemically coupled SPR sensing applications: (**a**) Heavy metal detection [36]. (**b**) Thrombin detection [37]. (**c**) Sialic acid detection [38].

**Figure 4 nanomaterials-13-02400-f004:**
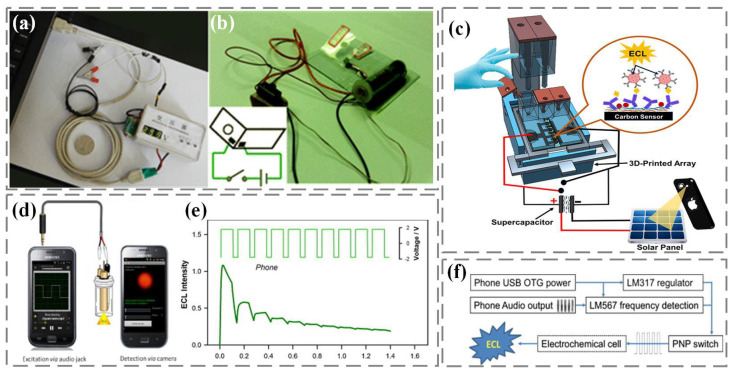
ECL excitation. (**a**) Rechargeable battery: lithium battery [76]. (**b**) Rechargeable battery: nickel–metal hydride battery [77]. (**c**) Supercapacitors [78]. (**d**,**e**) Smartphone audio port [79]. (**f**) Smartphone USB-OTG [80].

**Figure 5 nanomaterials-13-02400-f005:**
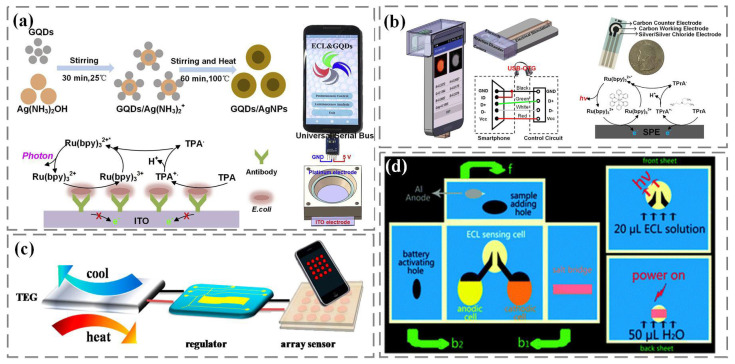
(**a**) Smartphone USB-OTG: nanocomposites and system [81]. (**b**) Smartphone-based electrochemiluminescence system with reaction chamber and electrodes [82]. (**c**) Thermal power supply [83]. (**d**) Self-powered 3D paper chip [84].

**Figure 6 nanomaterials-13-02400-f006:**
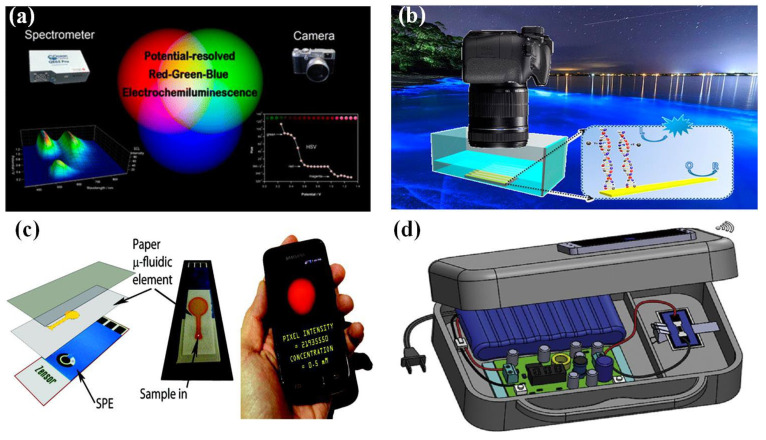
Luminescent detector: (**a**) Red-green-blue ECL using a digital camera as the detector [86]. (**b**) Digital camera visualization with a radio luminescence array [87]. (**c**) Paper-based microfluidics electrodes and smartphone-based ECL detection platforms [88]. (**d**) Handheld ECL detection platform [89].

## Data Availability

Not applicable.

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
