# Peer review of "Nanobiosensing Based on Electro-Optically Modulated Technology"

_nanomaterials, 2023, doi:10.3390/nano13172400_

Round 1

Reviewer 1 Report

S. Li et al. comprehensively reviewed nanobiosensing technology based on electrochemistry coupled with surface plasmon resonance and electrochemiluminescence. The basic mechanisms of the two phenomena are briefly introduced, and relevant applications are discussed in detail. Lastly, the authors provide an outlook and perspectives on these technologies, suggesting interesting ideas for future work. Overall, this manuscript offers an interesting overview of nanobiosensing based on electro-optically modulated technologies, which could be beneficial for readers from various fields such as nanophotonics, biochemistry, and biomedical engineering. However, I encountered some difficulty in reading the paper in terms of language. Thus, I recommend professional proofreading or improving the paper's quality before publication.

Capital letters appear in weird positions.

1.       In line 493, 'local Surface plasmon resonance' should be 'local surface plasmon resonance,' as stated in line 21.

2.       In line 43, 'Formant' should be 'formant.'

3.       In line 54, 'Analytical' should be 'analytic.'

4.       In line 59, 'Represents' should be 'represents.'

5.       In line 60, 'χ Determined by particle shape' should be 'χ is determined by particle shape.'

6.       In line 155, 'Cyclic' should be 'cyclic.'

Some words are abbreviated, and some are not.

1.       In line 492, 'Surface plasmon resonance' is abbreviated as 'SPR,' which is not used in line 493 and 494. Moreover, it should be abbreviated at the earliest possible sentence, not at the end of the paper.

2.       Words frequently used are better abbreviated at the earliest possible sentence. Once abbreviated, the same word should be presented in the abbreviated form throughout the paper. The following is the list of frequently used words: surface plasmon resonance, electrochemiluminescence, electrochemistry coupled surface plasmon resonance, gold, silver, bovine serum albumin.

3.       In line 263, triplet state-triplet state annihilation is abbreviated as TTA, which is never used afterward. In this case, you do not need to abbreviate it. The same applies to all other abbreviated words.

4.       Some sentences are not grammatically complete, for instance, from line 469 to 488. I could not catch all the grammatical mistakes; hence I suggest a careful reading and revision."

Overall, the revised version should address the mentioned issues and enhance the clarity of the text. However, please note that it was challenging to identify all potential grammatical errors or inconsistencies. A thorough proofreading of the entire paper will ensure its quality and readability.

Author Response

Response to Reviewer 1 Comments

We have revised our manuscript entitled “Nanobiosensing based on electro-optically modulated technology” (Manuscript ID: nanomaterials-2554294).

Thank you very much for reviewing our original manuscript and giving valuable comments and detailed suggestions. We have carefully reviewed our manuscript and conducted a comprehensive grammar check, including modifications to capital letters and abbreviations. We have also reorganized and clarified the image data, adding detailed descriptions of the cited content, making the manuscript easier to read.

Revised portions are marked up using the “Track Changes” function. The main corrections in the paper and the responds to the reviewer’s comments are as following.

If you have any question, please do not hesitate to let me know as soon as possible.

Sincerely yours,

Shuang Li

Prof. Ph.D.

Academy of Medical Engineering and Translational Medicine, Medical College,

Tianjin University, Tianjin, China

Tel and Fax: +86 022 83612122

  1. Li et al. comprehensively reviewed nanobiosensing technology based on electrochemistry coupled with surface plasmon resonance and electrochemiluminescence. The basic mechanisms of the two phenomena are briefly introduced, and relevant applications are discussed in detail. Lastly, the authors provide an outlook and perspectives on these technologies, suggesting interesting ideas for future work. Overall, this manuscript offers an interesting overview of nanobiosensing based on electro-optically modulated technologies, which could be beneficial for readers from various fields such as nanophotonics, biochemistry, and biomedical engineering. However, I encountered some difficulty in reading the paper in terms of language. Thus, I recommend professional proofreading or improving the paper's quality before publication.

Capital letters appear in weird positions.

  1. In line 493, 'local Surface plasmon resonance' should be 'local surface plasmon resonance,' as stated in line 21.
  2. In line 43, 'Formant' should be 'formant.'
  3. In line 54, 'Analytical' should be 'analytic.'
  4. In line 59, 'Represents' should be 'represents.'
  5. In line 60, 'χ Determined by particle shape' should be 'χ is determined by particle shape.'
  6. n line 155, 'Cyclic' should be 'cyclic.'

Response:

Thank you for your suggestion. We have thoroughly reviewed the incorrect capital letters in the manuscript and made modifications.

Some words are abbreviated, and some are not.

  1. In line 492, 'Surface plasmon resonance' is abbreviated as 'SPR,' which is not used in line 493 and 494. Moreover, it should be abbreviated at the earliest possible sentence, not at the end of the paper.
  2. Words frequently used are better abbreviated at the earliest possible sentence. Once abbreviated, the same word should be presented in the abbreviated form throughout the paper. The following is the list of frequently used words: surface plasmon resonance, electrochemiluminescence, electrochemistry coupled surface plasmon resonance, gold, silver, bovine serum albumin.
  3. In line 263, triplet state-triplet state annihilation is abbreviated as TTA, which is never used afterward. In this case, you do not need to abbreviate it. The same applies to all other abbreviated words.
  4. Some sentences are not grammatically complete, for instance, from line 469 to 488. I could not catch all the grammatical mistakes; hence I suggest a careful reading and revision."

Response:

Thank you for your suggestion. We have carefully reviewed the full text and revised the use of abbreviations.

Overall, the revised version should address the mentioned issues and enhance the clarity of the text. However, please note that it was challenging to identify all potential grammatical errors or inconsistencies. A thorough proofreading of the entire paper will ensure its quality and readability.

Response:

Thank you for your suggestion. We have thoroughly reviewed the grammar errors and made modifications, reorganized and clarified the image data, and added detailed descriptions and discussions of the cited data, making the manuscript more comprehensive and easy to read.

Reviewer 2 Report

In this Review, authors reported about s the basic concepts of LSPR, electrochemically coupled local Surface plasmon resonance, electrochemiluminescence and related biosensor detection applications.   Manuscript is appropriate for this journal, and I would recommend this work for publication, but after major revisions:

1) All images have a very poor quality! Authors have to find and insert pictures with appropriate quality

2) Figure 2 - each blocks of a) and b) consist of two figures. For instance, a) -current and transmission spectra. The graph, with transmission spectra, does not have a label. For the graphs without labels, they should be added.

3) Figures 2 and 3 are too cluttered with data, that is way, the figures are not convenient to read. I would recommend divide each of these figures on two parts and make instead of two figures four

4) The main remark - the description of the figures and their corresponding works is looks like the introduction to the some manuscript, but not a real Review. There is no minimum description of the submitted works: what kind and how the structures were made, what kind parameters were used in measurement, etc. Authors make a link to a picture with a poor quality and send the reader to read the manuscript from which it was adapted. For instance - lines 407-409 "In addition, Khoshfetrat et al. used digital cameras to visualize electrochemiluminescence signals and developed a radio iluminescence DNA array for visual genotyping of different single-nucleotide polymorphism (Fig. 4b) [87]." Еach work used in the review should be described in more detail, and not in general terms!

Author Response

Response to Reviewer 2 Comments

We have revised our manuscript entitled “Nanobiosensing based on electro-optically modulated technology” (Manuscript ID: nanomaterials-2554294).

Thank you very much for reviewing our original manuscript and giving valuable comments and detailed suggestions. We have carefully reviewed our manuscript, thoroughly reviewed the incorrect grammar and made revisions. We have downloaded all original images and distributed them reasonably according to the content of the manuscript, added image labels, clarified the images, and added detailed descriptions of the cited data to make the manuscript easier to read.

Revised portions are marked up using the “Track Changes” function. The main corrections in the paper and the responds to the reviewer’s comments are as following.

If you have any question, please do not hesitate to let me know as soon as possible.

Sincerely yours,

Shuang Li

Prof. Ph.D.

Academy of Medical Engineering and Translational Medicine, Medical College,

Tianjin University, Tianjin, China

Tel and Fax: +86 022 83612122

In this Review, authors reported about s the basic concepts of LSPR, electrochemically coupled local Surface plasmon resonance, electrochemiluminescence and related biosensor detection applications.   Manuscript is appropriate for this journal, and I would recommend this work for publication, but after major revisions:

  • All images have a very poor quality! Authors have to find and insert pictures with appropriate quality

Response 1:

Thank you for your suggestion. We have re-downloaded all the original data maps and processed the image clarity.

  • Figure 2 - each blocks of a) and b) consist of two figures. For instance, a) -current and transmission spectra. The graph, with transmission spectra, does not have a label. For the graphs without labels, they should be added.

Response 2:

Thank you for your suggestion. We have made modifications to all unlabeled graphics and added appropriate labels, and rearranged the image data to make the images clearer and easier to read.

Relevant description in “Sensing application” (In line 171, Figure 2a and 2b; In line 368, Figure 4b and 4e).

  • Figures 2 and 3 are too cluttered with data, that is way, the figures are not convenient to read. I would recommend divide each of these figures on two parts and make instead of two figures four

Response 3:

Thank you for your suggestion. We have rearranged the data in Figures 2 and 3 based on the manuscript content, dividing them into four images, and downloaded and processed the original images to make them clearer.

Relevant description in “Sensing application” (In line 171, Figure 2 and in line 210, Figure 3)

Relevant description in “(1) ECL excitation” (In line 368, Figure 4 and in line 394, Figure 5)

  • The main remark - the description of the figures and their corresponding works is looks like the introduction to the some manuscript, but not a real Review. There is no minimum description of the submitted works: what kind and how the structures were made, what kind parameters were used in measurement, etc. Authors make a link to a picture with a poor quality and send the reader to read the manuscript from which it was adapted. For instance - lines 407-409 "In addition, Khoshfetrat et al. used digital cameras to visualize electrochemiluminescence signals and developed a radio iluminescence DNA array for visual genotyping of different single-nucleotide polymorphism (Fig. 4b) [87]." Еach work used in the review should be described in more detail, and not in general terms!

Response 4:

Thank you for your suggestion. We have carefully reviewed our manuscript and added detailed descriptions to the simple citations in the manuscript, including the methods, principles, and performance parameters of the developed sensors. All images have been clearly processed and distributed according to the logical content of the manuscript, making it easier to read.

Relevant description in “Sensing application” (In lines 152-155 and in lines 158-161);

Relevant description in “(1) ECL excitation” (In lines 358-360, in lines 409-416 and in lines 428-439)

Round 2

Reviewer 2 Report

All responses to comments were done

Author Response

Thank you for your approval of our revised manuscript.